# An Experimental Framework for Developing Point-of-Need Biosensors: Connecting Bio-Layer Interferometry and Electrochemical Impedance Spectroscopy

**DOI:** 10.3390/bios12110938

**Published:** 2022-10-29

**Authors:** Sadia Fida Ullah, Geisianny Moreira, Shoumen Palit Austin Datta, Eric McLamore, Diana Vanegas

**Affiliations:** 1Division of Glycoscience, Department of Chemistry, School of Engineering Sciences in Chemistry, Biotechnology and Health, KTH Royal Institute of Technology, 106 91 Stockholm, Sweden; 2Department of Environmental Engineering and Earth Sciences, Clemson University, Clemson, SC 29634, USA; 3Global Alliance for Rapid Diagnostics, Michigan State University, East Lancing, MI 48824, USA; 4MIT Auto-ID Labs, Department of Mechanical Engineering, Massachusetts Institute of Technology, 77 Massachusetts Ave, Cambridge, MA 02139, USA; 5Medical Device (MDPnP) Interoperability and Cybersecurity Labs, Biomedical Engineering Program, Deparment of Anesthesiology, Massachusetts General Hospital, Harvard Medical School, 65 Landsdowne Street, Cambridge, MA 02139, USA; 6Agricultural Sciences, Clemson University, 821 McMillan Rd, Clemson, SC 29631, USA; 7Interdisciplinary Group for Biotechnology Innovation and Ecosocial Change-BioNovo, Universidad del Valle, Cali 76001, Colombia

**Keywords:** biosensor design, protein–protein interaction, molecular affinity, binding kinetics, analytical sensing, SARS-CoV-2

## Abstract

Biolayer interferometry (BLI) is a well-established laboratory technique for studying biomolecular interactions important for applications such as drug development. Currently, there are interesting opportunities for expanding the use of BLI in other fields, including the development of rapid diagnostic tools. To date, there are no detailed frameworks for implementing BLI in target-recognition studies that are pivotal for developing point-of-need biosensors. Here, we attempt to bridge these domains by providing a framework that connects output(s) of molecular interaction studies with key performance indicators used in the development of point-of-need biosensors. First, we briefly review the governing theory for protein-ligand interactions, and we then summarize the approach for real-time kinetic quantification using various techniques. The 2020 PRISMA guideline was used for all governing theory reviews and meta-analyses. Using the information from the meta-analysis, we introduce an experimental framework for connecting outcomes from BLI experiments (*K*_D_, *k*_on_, *k*_off_) with electrochemical (capacitive) biosensor design. As a first step in the development of a larger framework, we specifically focus on mapping BLI outcomes to five biosensor key performance indicators (sensitivity, selectivity, response time, hysteresis, operating range). The applicability of our framework was demonstrated in a study of case based on published literature related to SARS-CoV-2 spike protein to show the development of a capacitive biosensor based on truncated angiotensin-converting enzyme 2 (ACE2) as the receptor. The case study focuses on non-specific binding and selectivity as research goals. The proposed framework proved to be an important first step toward modeling/simulation efforts that map molecular interactions to sensor design.

## 1. Introduction

Understanding biomolecular interactions is critical for biotechnology development and have diverse applications. Examples include biotherapeutic drug and vaccine development [1] and biosensors [2,3]. A wide variety of techniques (both experimental and in silico) have been developed for studying the biomolecular complex that forms between a receptor and target. Data may be used to reveal information on binding mechanism, complex stability, affinity, specificity, of interaction kinetics.

Currently, the most common experimental instruments for characterization of biomolecular interactions are surface plasmon resonance (SPR) or bio-layer interferometry (BLI). These techniques allow real-time monitoring of binding events without the addition of exogenous labeling molecules. High-throughput modern instruments support parallel sample analysis with low sample volume, thus facilitating testing of many different experimental designs in multi-well plates [4,5]. When combined with in silico predictions of biomolecular interactions [6,7,8], the combination of experimental testing and simulation represents the state of the art in bioanalytical technology development. Ideally, this rational design approach would be the basis of biosensor design, informing the process from molecular recognition and transduction to signal acquisition and device application.

The operating mechanism of biosensor devices is based on molecular recognition between an immobilized receptor (also called a biorecognition element) and a target analyte associated with the sample [9,10]. The number of material combinations for biosensor development are nearly infinite, but the most common receptors include tethered or encapsulated proteins, nucleic acids, or whole microbial cells [11,12,13,14,15]. In principle, biosensor performance metrics depend on the kinetics of molecular interaction events occurring at the sensor-sample interface, but the nature of this dependence is not clear [16]. Rational assembling of biosensors based on computer-aided design are emerging [17,18], but this approach has been limited to studies that focus only on recombinant receptor design/synthesis. Particularly when a new material is used as the biosensor receptor (recombinant or other), a lack of detailed interaction data (experimental and/or in silico data) can be a bottleneck for the development of biosensors. Even when interaction data is available, the process of connecting high-throughput laboratory data and/or simulation data with biosensor development is not trivial. Thus, there is a critical need for approaches to connect experimental characterization of molecular interactions with biosensor key performance indicators (Figure 1).

Here, we briefly review models (Section 3), and practices (Section 4) that study biomolecular interactions relevant to biosensors. We then introduce the first layer of a framework for connecting the experimental design from these two domains (Section 5) and suggest a process composed of eight-steps. The purpose of the framework is to help biosensor developers choose from a vast library of biorecognition elements and transducers that are available for creating biosensor tools. The proposed framework applies bio-layer interferometry technology to select biorecognition and transducer combinations in sensor development. We provide a case study (Section 6) that focuses on application of the experimental framework for development of a SARS-CoV-2 biosensor.

## 2. Methods

The 2020 PRISMA (preferred reporting items for systematic reviews and meta-analyses) guideline was used for a meta-literature review on the governing theory and instruments covered in this review [19]. The PRISMA checklist was used to ensure study coherence; the PRISMA flow diagram was used to report the outcome of the analysis (Figure 2). For the primary literature meta-analysis, 131 records were identified, and reduced to 63 manuscripts and two websites that were used. In the case study, a meta-analysis with 720 records was screened and 39 records were used in the final analysis (see Section 6 and Appendix A for details of case study). In total, 851 records were identified and 104 were used in the study (88% screening rate).

### 2.1. Information Sources and Search Strategy

The information sources for meta-analyses in this work include three databases (Web of Science, SCOPUS, Google scholar) and one reference manager (Mendeley). In addition, website search (Google, Bing) was used to identify potentially relevant information. For all bibliographic sources, keyword search included the following: molecular interaction, biomolecular interaction, binding kinetics, molecular affinity, affinity, avidity, biosensor, electrochemical biosensors, biosensor design, biorecognition, transduction, protein–protein interaction, ligand-analyte interaction, and biotechnology.

### 2.2. Elegibility and Selection Process

Web of Science and SCOPUS: Following keyword search, records were pre-screened using search engine sort features (date, relevance, times cited) and then organized into auto-bin categories. Abstracts and article information from the top 20% of the auto-bin records were exported to Microsoft Excel for further analysis. All other manuscripts were removed prior to screening. For archiving files that did not meet the inclusion criteria of this work, the category grouping tool was used. When this tool was not available, authors independently screened each record and retrieved documents based on relevance.

Mendeley: After keyword search, records were pre-screened using search engine sort features (date within last 10 years, document type limited to journal and books, organized by relevance, times cited). A Mendeley folder was created for organizing the records, and then abstracts and article information from the top 20% of the auto-bin records were exported to Microsoft Excel for further analysis. All other manuscripts were removed.

Google Scholar and Bing: After keyword search, records (excluding patents) were pre-screened by date (within last 10 years), and further sorted by relevance. The library function was used to organize the records, and then abstracts and article information from the top 20% of the auto-bin records were exported to Microsoft Excel for further analysis. All other manuscripts were removed.

Records were combined into a single Excel Spreadsheet and screened for duplicates. Duplicates were identified using the sort function in Excel, and deleted using a Macro.

### 2.3. Screening and Synthesis

The title of each article which passes pre-screening was analyzed. Articles which had descriptive titles that did not fall within the scope of the literature review were archived. The remaining manuscripts were sought for retrieval. A fraction of articles were behind a pay wall and inaccessible. Inter-library loans were requested, and if not available the records were excluded. For records which fulfilled the screening criteria, an excel worksheet was created which organized each item by associated keyword(s). The sort function was used to organize the repository by various combination of keyword(s). Any record which contained missing or unclear information was assumed to be invalid and removed from the study. To limit bias, record location was conducted independently by each author, and the results of the process shared during manuscript preparation.

### 2.4. Proof-of-Concept

To demonstrate the usability of the proposed framework, data obtained from a recently published study on the development of an electrochemical ACE2 biosensor [20] was analyzed against the framework. The study used bio-layer interferometry as a qualitative screening tool to check binding interactions between truncated Angiotensin-Converting Enzyme II (ACE2) from a commercial supplier and recombinant Spike proteins as well as whole attenuated SARS-CoV-2 viral particles. Binding signatures generated from BLI outputs were used to access qualitative information about the selectivity of the ligand with target and non-target viruses. Finally, the BLI output was applied on the development of an impedimetric biosensor based on the same buffer conditions and coupling method approach.

## 3. Biomolecular Interactions: A Brief Review of Models Based on the Ligand-Analyte Complex

Biomolecular structure and size play a fundamental role in functions. Biomolecules may interact in reversible and irreversible binding events that perform a variety of functions. Biomolecule interactions that are relevant to biosensors include protein–protein, protein-peptide, carbohydrate-protein, enzyme-substrate, or DNA-protein events [21]. Molecular recognition is the basis for biosensor design, where both specificity and affinity between ligand and analyte are required for formation of a specific complex [22].

Numerous conceptual and mathematical models have described ligand-analyte interactions under the chemistry triplet [23,24,25]. Binding may be described by the symbolic representation (equation) or particulate representation (cartoons) shown in Figure 3A. As an example of protein–protein interaction, the particulate representation uses a generic monomer as receptor (ligand) and truncated epitope (target analyte). In this representation, binding (complex formation) is represented as the blue halo in the right box. In this scheme, target analyte [A] interacts with the receptor [B] to create an analyte-receptor complex [AB]. The stability of the complex [AB] is dependent on the thermodynamics of the system and the rates of association (*k*_on_) and disassociation (*k*_off_). An equilibrium interaction is established when the concentrations of [A], [B], and [AB] do not change according to the mass action law [26]. Given the state of quasi-equilibrium between these species, the interaction may be modeled mathematically using saturation kinetics, where the equilibrium constant (*K*_D_) is the concentration ratio of the involved species. Analogously, *K*_D_ can be defined as the ratio of *k*_off_ and *k*_on_, which is experimentally determined at the point in the titration regime equal to 50% of the maximum association rate. In order to detect and/or quantify stable binding, in a meaningful biological system, the association rate (*k*_on_) must be higher than the disassociation rate (*k*_off_).

When modeling kinetic processes, it is important to note which steps are rate limiting and the conditions under which the limitations are relevant. In the simplest case of binding, as depicted in Figure 3A, rate limitations are governed by solution temperature (Arrhenius behavior), solution pH and osmotic gradients, and occupancy of the binding site. In more complex interactions, cooperativity between molecules competing for the binding site alters the rate(s) considerably. This cooperativity (including allosteric mechanisms) may promote (positive cooperativity) or deter (negative cooperativity) subsequent binding of analyte molecules. Other multitopic interactions may include other complexities, such as surface charge phenomena (e.g., homobivalency or heterobivalency). Taken together, these interactions may be represented by the global *K*_D_, which is an approximation of the equilibrium between disassociation and association rates. The terms affinity and avidity are used (sometimes interchangeably) to describe the interaction between ligand and target analyte, where avidity is an increased association that may be due to one or more of the aforementioned rate-limiting factors (temperature, osmotic gradient, multivalency, etc.).

For binding kinetics described in Figure 3A, *k*_on_ is assumed to be significantly faster than *k*_off_ for interactions displaying affinity, but it is imperative that *k*_off_ is not assumed to be zero. In other words, the quasi-equilibrium between ligand and analyte does not imply the system is static, in fact disassociation is a necessity for characterization of binding kinetics. The approach allows for slow off-rates (*k*_off_) but does not specify that a non-reversible (i.e., covalent) bond forms between ligand and analyte. Interaction between ligand and analyte are dynamic. The time required for analyte disassociation is an important indicator of the analyte-ligand complex stability. In the disassociation phase, both analyte and ligand return to their original state (i.e., no conformational changes occur as a result of the analyte-ligand interaction).

Enzymatic catalysis shown in Figure 3B is an extension of the concept shown above, and despite the mechanism of binding/reaction in the glucose biosensor. In Figure 3B, the particulate representation depicts oxidation of β-D-glucose (red halo) to gluconolactone (green halo) and hydrogen peroxide (yellow halo) by the enzyme glucose oxidase (GOx). The symbolic states indicate that the interaction between analyte [A] and ligand [B] leads to formation of an intermediate ligand-analyte complex [AB] as in the previous example, but the process is extended. An extra post-binding step is included to account for the reaction that occurs after association. Like all reactions, this step requires an input of energy to initiate the process, known as the activation energy. Symbolically, the complex [AB] undergoes a reaction at rate (*k*_+__Rxn_), which alters the nature of the molecule, in this example creating a distinct product [P] that is no longer thermodynamically stable in the binding site of the enzyme. Given the local instability of the molecules, the product [P] disassociates and subsequently diffuses away from the binding site of the enzyme (referred to as the binding pocket). The binding site of the enzyme [B] is thus unoccupied and able to undergo subsequent cycles of this process with additional analyte molecules. To be comprehensive, the reverse reaction rate (*k*_-__Rxn_) is also shown. While it is thermodynamically reasonable to include the reverse reaction rate in the analysis of many oxidoreductase enzymes, we don’t discuss it here.

Michaelis-Menten (M-M) models this behavior (mathematically) based on the quasi-equilibrium that occurs between species. In order for this theorem to be valid, a few key assumptions are necessary. When viewed as a system of reactions, the association rate (*k*_on_) must be orders of magnitude higher than the disassociation rate (*k*_off_), as was true in the affinity discussion in Figure 3A. In addition, the associate rate is assumed to be orders of magnitude faster than the reaction rate (*k*_+__Rxn_). This would result in a non-diminishing pool of [AB], implying that the net reaction rate (*V*_0_) may be approximated as *k*_+__Rxn_*[AB] (note that this assumes that the reverse reaction rate, *k*_-__Rxn_, is negligible). This assumption also implies that at high concentrations of analyte [A], all of the binding sites will be occupied (as [AB]), and the resulting maximum net reaction rate (*V*_max_) is equal to the reaction rate (*k*_+__Rxn_*) multiplied by the total analyte in the system, bound and unbound ([A] + [AB]). Another key assumption for the Michaelis-Menten abstraction is that at early time points, the concentration of product [P] is negligible.

For catalysis kinetics, derivation of M-M kinetics (Figure 3B) assumes *k*_off_ is negligible (near zero), which contradicts the assumption for binding kinetics shown in panel A (Figure 3A). These models employ the free ligand approximation, which postulates that the target analyte concentration in suspension at any time is equal to the total ligand concentration in the system. If this postulate is violated (e.g., when the constant *K*_M_ is lower than the total enzyme concentration), other equations must be used (such as the Morrison equation [29]). In addition, reaction kinetics also assume that *k*_on_ is significantly higher than the forward reaction rate (*k*_+__Rxn_). This condition ensures an infinite pool of complex [AB] is perpetually available for the reaction to occur. Finally, the forward reaction rate (*k*_+__Rxn_) is expected to be orders of magnitude faster than the reverse reaction rate (*k*_-Rxn_), which drives the reaction to the right of the equation shown in panel B (Figure 3B); thus, implying continuous depletion of analyte and formation of product, while the concentration of the bioreceptor (i.e., catalyst) remains constant throughout the process. All of these premises are critical for the kinetics framework shown in panel B (Figure 3B), but the key difference with the binding kinetics in panel A (Figure 3A) is the mathematical treatment of *k*_off_.

Neither the binding kinetics in panel A nor the M-M kinetics in panel B are directly applicable for allosteric regulation. The most common allosteric models for binding kinetics are expansions of the classic Hill model [30] and Hill-Langmuir model [31]. Allosteric models for reaction kinetics include the concerted (MWC) model [32] and the sequential model [33], among others. It is useful to recognize the elegance of the mathematics in these models, as well as the diverse uses in myriad of applications. Although allosteric models may be broadly categorized under cooperative models, care must be taken to ensure that the assumptions of any model are specifically relevant to the particular study.

Given the assumptions above, the rate-limiting step for enzymatic catalysis is the conversion of analyte to product in the catalysis step (from complex [AB] to [P] + [B]). The presence of inhibitors or insufficient activators (i.e., cofactors) can exacerbate this rate-limiting condition, which leads to a decrease in the catalytic efficiency. In this ideal generic system, conditions that are non-ideal for binding are neglected (e.g., decreased temperature, non-ideal pH or osmotic gradient, negative cooperativity). The seminal work by Michaelis and Menten is still used as a foundation of models today, but emerging adaptations address the aforementioned non-idealities, as well as other compounding affects (e.g., complex regulatory modalities) [34,35]. While the calculation of rate constants for both binding (Figure 3A) and enzymatic reaction (Figure 3B) are each based on the general mathematical framework of the rectangular hyperbola, the equations are not interchangeable. Some of the assumptions made during derivation of each model system are incompatible.

These fundamental models for characterizing biomolecule interactions are central to the first step in a biosensor process (molecular binding and complex formation). In the next section, we briefly review experimental techniques for quantifying binding interactions. We focus attention on one modern techniques that provide high throughput analysis, which are increasingly gaining popularity in biosensor research (biolayer interferometry).

## 4. Methods for Characterizing Ligand-Analyte Interactions

As reviewed in Section 3, protein structure and substrate binding are interwoven. Many experimental methods are available to characterize ligand-analyte interactions. Common techniques for assessing protein structure include X-ray crystallography, Laue X-ray diffraction, small-angle X-ray scattering (SAXS), nuclear magnetic resonance (NMR), and cryo-electron microscopy. In addition to these experimental techniques, in silico simulation of ligand-analyte relationships are powerful for informing experimental work, and *vice versa*. Use of computer simulations for studying biomolecule interplay has been reviewed extensively [36,37,38,39,40].

In addition to structural characterization (experimental or in silico), thermodynamic techniques are also used to predict ligand-analyte behavior [41]. Isothermal titration calorimetry (ITC) and/or differential scanning calorimetry (DSC) are commonly used for quantitative thermodynamic analysis. The outcome of ITC studies is an overview of the global thermodynamic parameters associated with the ligand-analyte coupling. DSC provides information on the biomolecular complex and any free component(s). Combination of ITC and DSC allows the global parameters from ITC to be decomposed into specific contributions related to conformational changes. Jelesarov and Bosshard [41] noted more than 20 years ago that ITC and DSC, if combined with in silico structural modeling, is one of the most powerful cluster of techniques of linked function analysis of biomolecular interaction (or any coupled equilibria system). Indeed, this combinatorial approach is the current state of the art for studying rigid body binding and/or structure–function relationships [41,42,43,44].

In this manuscript, we focus on experimental analysis of ligand-analyte interactions. The most common techniques include surface plasmon resonance (SPR) [45] and localized surface plasmon resonance (LSPR) [46,47], fluorescence polarization (FP) [48,49], grating coupled interferometry (GCI) [50], bio-layer interferometry (BLI) [51,52,53] (Table 1). SPR is capable of quantifying kinetics in the range of 10^−9^ to 10^−3^ M [54] and can also provide insight into the binding enthalpy via van’t Hoff analysis [55]. FP is an equilibrium method based on determination of IC50 and is capable of simultaneous characterization the thermodynamics and kinetics of protein-ligand interactions. FP typically employs competition assays to displace labeled ligand molecules [56,57,58], and is a simple and low-cost assay. GCI is a hybrid phase-shifting Mach–Zehnder interferometer which has reduced noise due to absence of motors or moving parts and time domain measurement. GCI is becoming popular for use in analysis of protein-ligand interactions as well as other sensor targets [59].

BLI is a label-free, real-time method for characterizing association and disassociation kinetics based on interferometric shift at the tip of a glass fiber probe. Numerous protocols for the BLI technique are available [60]. BLI does not require fluidics or control systems and is a relatively simple multiplexing system. The technique appears insensitive to matrix pH or changes in refractive index (i.e., high solvent tolerance) [51,53,61]. In addition to protein–protein studies [51,62], BLI is also used for other binding system’s aptamer-target interactions [60,62,63,64].

**Table 1 biosensors-12-00938-t001:** Comparison of analytical experimental methods for characterization of ligand-analyte interactions.

Analytical Laboratory Method	Method Principle	Features	Reference(s)
Surface plasmon resonance (SPR)	Flow-based system. Measure changes in the refractive index near a chip-sensor surface. Ligand molecule is immobilized on chip-sensor surface. Analyte molecule is injected into an aqueous solution as a continuous flow cell.	Real-time, label-free, high-throughput, quantification of binding kinetics in flow through system	[46,47,65]
Biolayer interferometry (BLI)	Optical dip-and-read system that measures interference patterns between waves of light on fiber-optic biosensor with immobilized ligand.	Real-time, label-free, high-throughput in microwell format	[66]
Fluorescence polarization (FP)	Fluorescent protein variant fused to one of the protein partners.	Real time, labelled fluorophore, typically in microwell format	[67,68]
Grating coupled interferometry (GCI)	Target protein immobilized onto specialized sensor chips and the passage of analytes over the chip surface are monitored as time-dependent changes in refractive index.	Real time, label-free, reliable kinetics quantification in flow through system	[50,69]
Isothermal titration calorimetry (ITC)	Microcalorimeter quantifies absorption or release of heat during gradual titration of the ligand into a sample cell containing the analyte of interest	Label-free, complex stability study, evaluation of thermodynamic parameters in a sample cell	[70]

The advantages and disadvantages of SPR, FP, GCI, and BLI have been proposed by instrument manufacturers [71,72]. One peer-reviewed study by Murali et al. [4] compared the application of SPR and BLI with advantages and limitations for whole virus-binding studies. This work summarized the principles of both SPR and BLI and the application of these robust methods for whole virus-based studies based on several case studies. In a study by Jecklin et al. [73], SPR and ITC were compared for binding studies focused on human carbonic anhydrase I. This work showed that for some ligands, the agreement between SPR and ITC was excellent, while it was poor for others. One of the biggest problems with FP compared to BLI or SPR is that FP uses an indirect response, whereas the aforementioned techniques are direct read-out [74]. FP is based on proportional binding (IC_50_), which can lead to bias in measured affinity values based on specific experimental conditions used. Weeramange et al. [16] analyzed the limitations of BLI for lactose repressor protein binding immobilized DNA and found that the amplitude of BLI signal at equilibrium below *K*_D_ was lower than anticipated, which suggests issues with low signal-to-noise ratio. BLI and SPR both suffer from the practical problem of high experimental variation, which is associated with manufacturing complications (specifically, biosensor tip variations amongst manufacturing batches). The techniques in Table 1, which are label-free and real-time (BLI, SPR, GCI), have a major advantage over other methods when using the data to design biosensors. The advantage lies in the ability to quantify association (*k*_on_) and disassociation (*k*_off_) rates, parameters which inform biosensor design when working in complex matrices. In this manuscript, we focus on BLI and develop a framework for connecting measured outcome(s) from real-time, label-free techniques such as BLI.

### 4.1. Biolayer Interferometry: Basic Principles

BLI is an optical label-free technique based on a dip and read format. Molecular interactions are measured by the analysis of interference patterns of white light reflected from the surface of a biosensor tip. The ligand molecule is immobilized on a coated tip (biolayer) to produce an optical biosensor. Next, the biosensor is dipped in a sample solution containing the target analyte, and wavelength shifts at the probe-sample interface are recorded [60]. Interferometric signal (reported as shift, nm) derives from the reflection of polychromatic light interacting with a layer of immobilized protein on the tip of a fiber optic sensor [66]. Typical sensorgrams are derived from interferometry plots, which are time series data that report interferometric shift (in nm) during interactions between ligand and analyte.

BLI produces chrono-interferometric plots for baseline drift, ligand loading, baseline equilibration, ligand-analyte association, and ligand-analyte disassociation. Association curves are used to calculate the half saturation rate (s^−1^), half-life (nM), and the association rate constant, or *k*_on_ (M^−1^ s^−1^). The disassociation time series plots are used to calculate the disassociation rate constant, or interaction half-life, *k*_off_ (s^−1^). Under most conditions (>0.01 *K*_D_), *k*_off_ is not dependent on analyte concentration. Thus, the half-life is equal to the disassociation constant *k*_off_. If the ligand of interest is known to be multivalent in nature, the *K*_D_ values must be reported as apparent *K*_D_ (*K*_D_app_). There are numerous techniques for exploring valency, such as calculation of the Hill number (n_H_) using the Hill-Langmuir-Adair theory.

The two most common experimental techniques for improving kinetic analysis are (i) modifying ligand loading and/or (ii) modifying analyte concentration. To improve data reliability, many BLI studies utilize two internal controls (referred to as double reference data): (i) one control sensor to correct for signal drift (a bare fiber with no receptor), and (ii) one SAV-coated fiber with no analyte binding to correct for non-specific binding (NSB) [52]. Data from controls are commonly subtracted from association and disassociation binding curves. Application of data smoothing may appear to improve precision but may also introduce errors and bias.

Figure 4 shows a representative (ideal) BLI sensorgram, indicating regions of ligand loading, analyte association, and disassociation. Sensograms typically follow the sequence shown in Figure 4, including a buffer baseline, ligand loading step, additional buffer baseline, association step, and finally, disassociation step. It’s important to note that Figure 4 depicts the deployment of Streptavidin fibers (SA biosensor), but there are dozens of other material combinations that could be used (for example, Nickel charged tris-NTA with binds to his-tagged proteins).

### 4.2. Bio-Layer Interferometry: Common Experimental Approach for Biosensor Development

The 96 microwell format of BLI (or in some cases 384 microwells) supports a wide range of testing (Figure 5). The experimental design begins with a baseline step where the fibers are immersed in a buffer solution. Next, the fibers dip in a loading solution with the ligand partner at a pre-determined concentration, generally in μg/mL. After ligand immobilization, fibers are dipped in buffer solution for a second baseline for equilibration, assess assay drift, and determine the loading level of ligand. Next, fibers are dipped in the ligand’s binding partners solution for analyte association step. For association step, a titration down of analyte is prepared to assess kinetics metrics (*k*_on_, *k*_off_, *K*_D_) which leads to useful information related to the sensitivity and limit of detection. Finally, following analyte association, fibers are dipped in a buffer solution for dissociation step where the bound analyte is allowed to come off the ligand partner. A titration regime of up to eight measurements can be performed in a 96 microwell per run. At least two references may be added in the experimental plan: (i) a reference sensor with no ligand to correct for signal drift, and (ii) a reference sample with no analyte to check nonspecific binding. A pre-hydration step of the fibers are required before running the experiments to decrease assay drift and background signal during the experiment. Pre-hydration should be performed in the same assay buffer solution for at least ten minutes. These example of BLI experimental plans produce outcomes which inform biosensor development, although numerous other experiments are also possible such as assays for hysteresis and operating range by looking at the titration and binding regime. Depending on the biomolecules involved in the interactions, an experimental plan may require specific conditions such as optimum pH or trace amount of ions. Due to the high-throughput nature of BLI equipment, it allows testing of experimental conditions to optimize the binding once the ligand-analyte partner has been selected. In addition, quantitative measurements of biomolecule associations can be performed to evaluate binding affinities [75]. Once the ligand-analyte partner and optimal binding conditions are selected, the detection can be simulated through electrochemical testing by the biosensor and then applied in a real-world sample and/or context.

Numerous studies have combined kinetic analysis with biosensor development with the aim of improving device performance through detailed understanding of molecular interactions [76,77,78,79]. However, to date, there is no framework for connecting the output from molecular interaction studies (e.g., on rate, off rate, affinity constant) with biosensor key performance indicators (KPI).

## 5. A Framework for Connecting Biomolecular Interaction Parameters with Biosensor Engineering

Traditionally, the rational design concept is applied to in silico design of new biomaterials and/or nanomaterials, followed by experimental verification using various techniques [80]. Some argue that directed evolution approaches, or combinations of these two approaches, are advantageous with respect to classic protein engineering [81,82,83,84]. Both of these frameworks have been extremely powerful for protein engineering and have also advanced the field of biosensor development. What is missing from the current literature is an extension of the framework for producing key experimental data to ensure the design features are robust in a variety of scenarios. In this section, we summarize each of the three areas shown in Figure 6, then propose a sequence of steps for the rational framework in biosensor design. For a summary of experimental interaction studies, see Section 4.

### 5.1. Structural Analysis

Numerous experimental and computational methods for biomolecule structural characterization are summarized in Section 4 (Table 1). If data are not available from existing databases (e.g., protein data base, apta-index, etc.), a combination of experimental and computational methods must be used to produce key information. The important information derived from structural analysis includes: (i) location and number of binding site(s), (ii) conditions during analysis (pH, temperature, salinity), (iii) predicted or known structural stability under planned application of biosensor (if known).

### 5.2. Biosensor Key Performance Indicators

Biosensor key performance indicators include analytical sensitivity, analytical selectivity, response time, limit of detection, and clinical performance (if applicable).

### 5.3. Step-by-Step Guide to Applying Framework

Figure 7 depicts the sequence of steps to applying our proposed framework for biosensor development. A complete step-by-step guide for applying the framework can be found in Appendix A.

In the following sections, we provide an example case study of the proposed framework based on Moreira et al. [20]. BLI was used to characterize the binding affinity between SARS-CoV-2 spike protein RBD (recombinant protein containing the receptor binding domain) and truncated ACE2 containing the spike protein recognition domain. By truncating ACE2 and only focusing on the RBD, the work by Moreira et al. [20] reduced the number of possible molecular interactions and focused only on analyte binding (i.e., TMPRSS2 is uninvolved). This approach simplified the behavior of ACE2 to the non-reactive biding model, as depicted in Figure 8, allowing us to explore the use of ACE2 as a functional component of a device for detecting SARS-CoV-2 [20]. In the next section, we apply the eight-step process for the framework developed herein to development of a capacitive biosensor.

## 6. Case Study: Application of the Framework for Development of a SARS-CoV-2 Biosensor

The eight-step process for rational biosensor design is discussed in each subsection below. Note that we limit the case study to one specific research question and the associated experimental plan. The literate and data collected do not represent the complete study that was published by Moreira et al. [20] (Appendix A).

### 6.1. Goal of Research

The research goal of the biosensor design shown here was to evaluate the suitability of ACE2 receptor as biorecognition element in an electrochemical SARS-CoV-2 detection approach.

### 6.2. Intended Use of Proposed Device

The intended use of the proposed biosensor was to facilitate in situ SARS-CoV-2 diagnosis via saliva testing within moderate-complexity settings (e.g., nursing homes).

### 6.3. Research Question(s)

In this manuscript, we highlight one key research question: How specific is the ACE2 biosensor when exposed to other RNA viruses, including variants of SARS-CoV-2?

### 6.4. Perform Meta-Analysis of Published Literature and In Silico Analysis

The literature analysis for the case study followed the same structure as described in the methods section, with a few key differences. The meta-analysis by Datta [85] was used as the initial library for the case study (755 total records). The eligibility and criteria process described above were applied to reduce the number of records (pre-screening). As described above, screening was based on review of title and abstract. Where necessary, preference was given to records which were from peer-reviewed journals, and other records (websites, media outlets, etc.) were excluded from the study (see Appendix A). The analysis resulted in 37 records (plus 2 websites) in the case study, including one research article describing the development of a biosensor [20].

The literature showed that angiotensin-converting enzyme-2 (ACE2) is a suitable candidate for developing a host membrane receptor biosensor. ACE2 is a metallocarboxyl peptidase and angiotensin is a crucial regulator of the renin-angiotensin-aldosterone system [86]. In virology, ACE2 has been identified as the first point of entry for host infection with SARS-CoV-2 [85], but the eventual fusion of the viral and host membranes involves a cascade of events. Subunit 1 (S1) of the spike protein on the SARS-CoV-2 capsid surface binds with ACE2 during initial interaction [87,88] (it is worth noting that this binding is unrelated to the physiological enzymatic function of ACE2 and hence must not be conceptualized as an enzyme-substrate binding). During the cellular infection with SARS-CoV-2, the binding step likely involves simultaneous interaction of two S-glycoprotein trimers to an ACE2 dimer [89,90]. Subsequently, spike protein subunit 1 and subunit 2 are cleaved by the protease TMPRSS2 and a cascade of events then leads to membrane fusion for cell entry [90,91].

In this work, we exclusively focus on the affinity of a protein for the analyte of interest (the anticipated reaction is entirely unrelated to the enzymatic function of the molecule in use). The modeling of ligand-analyte interactions performed by in silico analysis is shown in Figure 8. The binding kinetics govern the interaction, and there is no downstream decrease in activation energy. The analyte of interest herein is the receptor binding domain (RBD) of the spike protein in the SARS-CoV-2 virus. The ligand is the truncated binding domain of the ACE2. This case study exclusively examined the binding affinity between SARS-CoV-2 spike protein RBD and truncated ACE2 containing recognition domain for the spike protein. By truncating ACE2 and only focusing on the RBD, we are experimentally reducing the number of possible molecular interactions and focusing only on analyte binding. This simplifies the expected in vivo behavior of ACE2 to the non-reactive binding model, as depicted in Figure 8, allowing us to explore the use of ACE2 as a functional component of a device for detecting SARS-CoV-2.

**Figure 8 biosensors-12-00938-f008:**
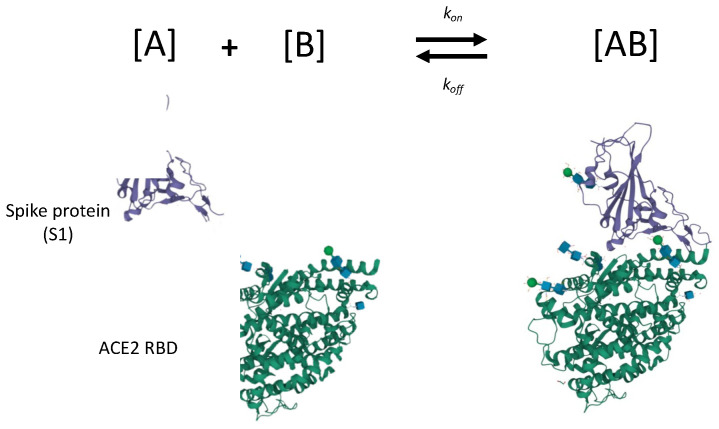
Schematic for binding of recombinant spike protein (subunit 1 core; recognition binding motif (RBM) not shown) and truncated ACE2 recognition binding domain (RBD). Each recombinant protein is truncated for experimental isolation of binding kinetics (e.g., enzymatic portion of ACE2 is cleaved). Most experimental studies focus on binding truncated ACE2 RBD with spike protein subunit 1 (shown here), and subunit 2, but few focus on inactivated SARS-CoV-2 virion. Crystal structures of spike protein subunit 1 and ACE2 are courtesy of Shang et al. [92]; program by Sehnal et al. [28] used for modification and export.

### 6.5. Molecular Interaction Study

In this study, we focused specifically on the ability of ACE2 to interact with the Spike protein and non-specific binding towards non-target molecules (Appendix A). Figure 9A shows representative BLI sensorgrams from interaction of biotinylated ACE2 and DELTA spike protein (SARS-CoV-2) in bicarbonate buffer at 30 °C. Rinse steps are not indicated in the plot. Another key experimental detail is the recording time, which must be sufficient for accurate quantification. In this representative example, recording time was eight times longer than the half-life (8 t), which represents 99.6% binding based on half-life theory. In general, the change in interferometric signal after ligand loading (0.5 to 0.7 nm) is significantly higher than the association step (0.1–0.2 nm). The signal from the association step is relatively low due to the reduction of frequency-modulated white light in the system.

Figure 9B shows a typical BLI curve (association and disassociation only, ligand loading and rinse steps not shown) for SARS-CoV-2 DELTA variant using [biot]ACE2 on SAV-coated fibers as the ligand in bicarbonate buffer at 30 °C (raw BLI data with no smoothing or internal controls). The precision of the data is low when comparing curves within the titration regime, in particular (see data between 350–500 s). However, differences in sensorgrams for analyte concentrations in the binding regime are clearly discernable for raw data. Signal smoothing (a *post hoc* analytical technique) is used to resolve the data in the titration regime and compared to double reference data (an empirical technique). The sensorgram in Figure 9B is similar to the expected values for protein–protein interactions [51,53,61]. The assay was selected as an example of an experiment that was conducted over concentrations that span the binding regime and the titration regime. The minimum concentration shown here (≈0.1*K_D_) represents a common low end of the dilution range, as analyte concentrations below this value typically do not show binding in other affinity studies [93]. Careful control of the experimental analyte concentration regime is critical for accurate determination of kinetic parameters [53]. In particular, experiments should include at least three data points within the binding regime for accurate quantification of binding kinetics (as shown here).

Many BLI kinetic analyses produce shift data for binding on the order of 0.1–0.2 nm, which is assumed to be a significant signal relative to the instrumental noise (reported to be 3.5 pm by the manufacturer [63]. In addition, BLI studies assume that temperature and shake speed (mass transfer) does not change during the experiment, but there are no onboard sensors [94,95] sensor to validate this assumption. The key points for Figure 9 are that: (i) the sensorgram indicates that binding and disassociation are taking place, and (ii) the recording time (8 t, which represents 99.6% binding based on half-life theory) is sufficient for capturing on/off kinetics. One caveat for BLI experiments of this type is the lack of published data studying the effect of mass transfer (shake speed) on binding kinetics at this level of granularity.

As demonstrated by Moreira et al. [20], BLI sensorgrams provide qualitative information about specificity of a ligand towards target analytes and non-target analytes (Figure 10). The magnitude of the response signal (wavelength shift, nm) could be a preliminary indication of biomolecular interactions or formation of the ligand-analyte complex. On other hand, the absence or low intensity of the response signal can give qualitative information about weak or absence of the ligand-analyte complex. This type of interpretation could be applied as preliminary indications of the selectivity and specificity of a given ligand for the development of technologies based on binding events.

In this case, study, BLI was designed to mainly evaluate ligand selectivity when in the presence of non-target analytes, composing the first part of the experimental design in our framework. Next, the receptor-target interaction was demonstrated qualitatively through representative association and dissociation curves (Figure 10) and quantitatively through *k*_on_/*k*_off_ maps (Appendix A), composing the second part of the framework. Finally, BLI outputs lead to device development, where sensitivity, response time, limit of detection, selectivity, and operation range were evaluated for the electrochemical sensing approach.

### 6.6. Biosensor Development

Electrochemical biosensors are analytical sensing tools able to transduce biochemical events in the recognition matrix (biosensor interface) to electrical signals. Within biochemical events are the classic antigen-antibody interaction and enzyme-substrate reaction. In addition to the classic biochemical events, any interaction at the molecular level between a biorecognition element and a target molecule can be transduced into an electrical signal by electrochemical biosensors. Several electrical signals, such as current, impedance, and voltage, can be generated depending on the transduction method [96]. The readout signal is proportional to the concentration of a target analyte in the sample, which allows the detection and quantification of a specific target in complex biological and environmental samples [21].

Electrochemistry-driven biosensing approach has attracted attention for allowing the miniaturization of laboratory analyses for on-site application. Electrochemical sensing approach for detection and/or measurement provides advantages compared with the laboratory settings approach. Electrochemical sensing could potentially be highly sensitive and selective towards a target analyte (a molecule or microorganism). In addition, electrochemical biosensors may be developed based on cost-affordable and environmentally friendly materials. Furthermore, electrochemical biosensors can be developed as a tiny device form and be deployed as point-of-care providing simple, fast, and accurate outputs on-site settings with a user-friendly interface.

As electrochemical biosensing is based on the interactions between a ligand-analyte partner in the recognition matrix, ideally, the first step in developing biosensors would be to analyze the nature of these interactions. Ligand-analyte affinity, optimal buffer binding conditions (pH, temperature, buffer components), and association and dissociation rates between ligand-analyte are crucial parameters for developing biosensors with high performance. In this context, BLI provides a high throughput screening for ligand-analyte and buffer conditions for electrochemical biosensor development. Data generated by BLI approach may help to define the best receptor for electrochemical sensing based on affinity constant (*K*_D_), association (*k*_on_), and dissociation (*k*_off_) rate. These metrics indicate the strength of ligand-analyte interactions. A higher affinity between ligand-analyte means a lower *K*_D_ value generated by BLI approach, and it is desired when the biosensor intends to detect a pathogen, for example. This means that a fast recognizing and tight binding between molecules occurs after exposure to an analyte (high *k*_on_). There is strong stability of formed complexes which slower dissociation or unbinding (low *k*_off_), which can generate a distinct electrical signal compared to a blank sample in the electrochemical testing. Development of biosensors based on high-affinity ligand-analyte partners can generate devices with high performance, such as high sensitivity and selectivity towards the target molecule with a lower limit of detection.

As demonstrated by Moreira et al. [20], BLI was used for qualitative interaction screening of hACE2 with target and non-target analytes to prove the selectivity and sensitivity of the receptor. Based on the same sensing format and conditions, an impedimetric ACE2-based biosensor was developed. Electrochemical sensing was selected as the transducer method, using a laser-induced graphene electrode. Streptavidin-biotin affinity was used as coupling method to attach ACE2 receptor to the electrode surface. Electrochemical impedance spectroscopy (EIS) was used for signal transduction. It’s important to note that the same experimental design (buffer conditions, streptavidin-biotin coupling method, target, and non-target analytes) was translated from BLI to electrochemical approach. A similar electrochemical output is depicts in Figure 11 for selectivity assay. As the electrochemical sensing field evolves, more emphasis is being placed on selectivity, hysteresis, operating range, and other key parameter indicators. The aforementioned parameters are largely driven by *k*_off_ and/or *k*_on_. The ACE2-based biosensor developed by Moreira et al. [20] showed an estimated limit of detection (LoD) of 2960 copies/mL, with a response time of less than 30 min.

In addition to its application in the study of molecular interactions, BLI and/or other techniques such as SPR can also be applied as centralized analytical biosensors [96,97,98] or design of advanced materials [99]. As a centralized analytical technique, BLI offers high sensitivity, response time around 10–20 min, and does not require extensively trained personnel. Among the limitations is the high-cost of equipment and optical biosensors tips, and the inability to deploy as point-of-need. Moreover, BLI has been used in the workflow of biosensor design for numerous aims, including fragment screening [100] and ligand-target design [101], for application in the drug discovery process and detection of the disease-specific biomarker.

BLI can provide a suitable understanding of the nature of biomolecular interactions, in quantitative (kinetics values) and qualitative (binding profiles) terms. However, the signal (wavelength shift, nm) generated by the BLI is directly proportional to the mass of the molecules (ligand and analyte). Interactions with small target molecules may not be detectable since a low molecular mass may not generate distinguishable changes in the biosensor tip. In addition, constant agitation is required to prevent re-binding between molecules, and the sample is subject to evaporation over time in BLI experiments. Electrochemistry-driven biosensing approach is sensitive to small changes on the electrode surface and has several mechanisms for electrical signal transduction. However, unlike BLI, electrochemistry approaches to biosensor regeneration are not common practice, as many electrochemical biosensors are designed for one-time use. For both approaches, the immobilization method may alter the conformation or orientation of the ligand, compromising binding events towards a target. A comparison between optical and electrochemistry approaches is shown in Table 2.

## 7. Conclusions and Future Perspectives

BLI provides sensitive, specific, qualitative, and quantitative information in real-time assays for screening biomolecule interactions. This review focused on applying BLI as the first step for electrochemical biosensor development. We proposed a framework to bridge bio-layer interferometry studies with electrochemical sensing attempting to connect output(s) of molecular interaction studies with key performance indicators used in the development of point-of-need biosensors. To date there is no unifying framework for the rational design of biosensors that is rooted in studies of binding kinetics, connecting output of interaction studies with input of biosensor design practices. In the next section, we extend the rational design framework to include this experimental approach. Based on defined study goal, research questions, and robust literature review, the framework proposed the development of an experimental design to study molecular interactions as first step for biosensor development. Qualitative (real-time sensorgrams) and quantitative (affinity constant, association and dissociation rates) data can be applied to select a receptor for the target analyte, which can provide a platform to compose biosensors. BLI allows screening and optimization of binding conditions to improve the electrochemical detection of binding events. Nonspecific binding is the major limitation in these techniques that requires an in-depth optimization and selection of immobilization methods to prevent nonspecific binding. As a study case for proof-of-concept of the proposed framework, we depicted an analysis of BLI sensorgrams for the interaction between the ACE2 protein with Spike protein of SARS-CoV-2. The complementarity between optical and electrochemistry approaches may generate high-performance and point-of-need deployment of analytical devices. Biosensors are feasible to use in public places and also in field to analyze the clinical and environmental samples. In addition, biosensors have a great significance for deployment in real-world contexts with wide application in fields such as agriculture/food safety, human health, environmental monitoring, and industry. Point-of-need biosensors rationalized in this review can offer a sustainable and repetitive diagnostic tool to apply to a wide variety of samples. Although not discussed here, this framework could be applied for many types of biorecognition (proteins, aptamers, peptides, etc.) or transducers (optical, electrochemical, magnetic, whole cell/organism). The aim of this manuscript is to provide a general platform that may be expanded upon in other studies.

## Figures and Tables

**Figure 1 biosensors-12-00938-f001:**
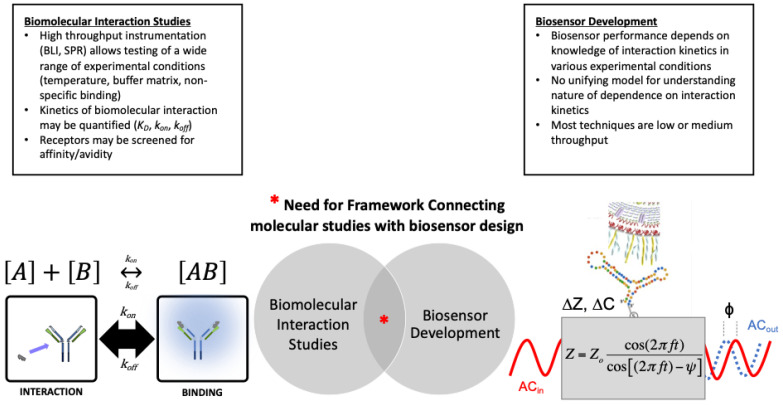
There is a need for frameworks which delineate the relationship between biomolecular interaction studies and biosensor development. High throughput instruments and in silico models for studying biomolecular interactions facilitate detailed characterization of receptor-target affinity under controlled conditions. Biosensor performance is based on knowledge of biomolecular interactions and kinetics, but have relatively low throughput. This article reviews key literature and proposes an experimental approach which may initiate development of a biosensor development framework. We propose one aspect of such a larger framework, focusing on experimental data and we establish an eight-step process. We apply this in a case study that focuses on SARS-CoV-2, utilizing biosensor development data collected from 2020–2022 in response to the COVID-19 pandemic.

**Figure 2 biosensors-12-00938-f002:**
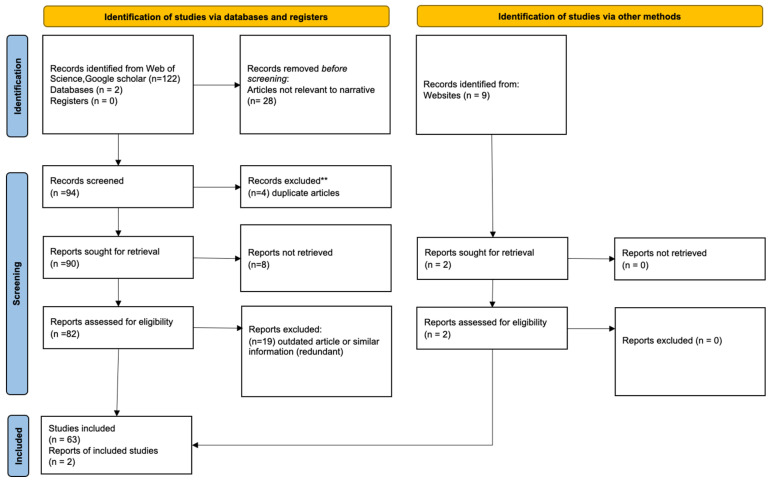
Meta-analysis focused on literature that discusses biomolecular interactions and/or biosensor design.

**Figure 3 biosensors-12-00938-f003:**
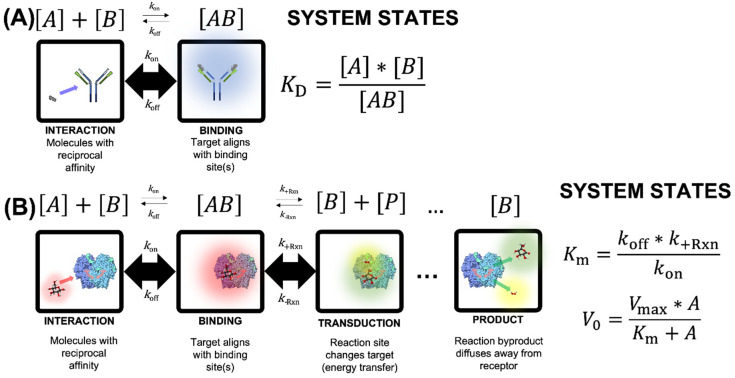
Symbolic and particulate schematics for (**A**) binding (top) and (**B**) binding + reaction (bottom). Although the derivation of the two model systems uses a similar approach, the mathematical underpinnings are quite different, and derivation is based on mutually exclusive assumptions. In panel A, the particulate representation uses a generic monomer as receptor (ligand) and truncated epitope (target). Binding (complex formation) is represented as the blue halo. In panel B, the particulate representation depicts oxidation of β-D-glucose (red halo) to gluconolactone (green halo) and hydrogen peroxide (yellow halo) by the enzyme glucose oxidase (Gox), oxygen (O_2_) is not shown. 3D structure is available in the Protein Data Base (PDB), according to Wohlfahrtet et al. [27] and Sehnal et al. [28].

**Figure 4 biosensors-12-00938-f004:**
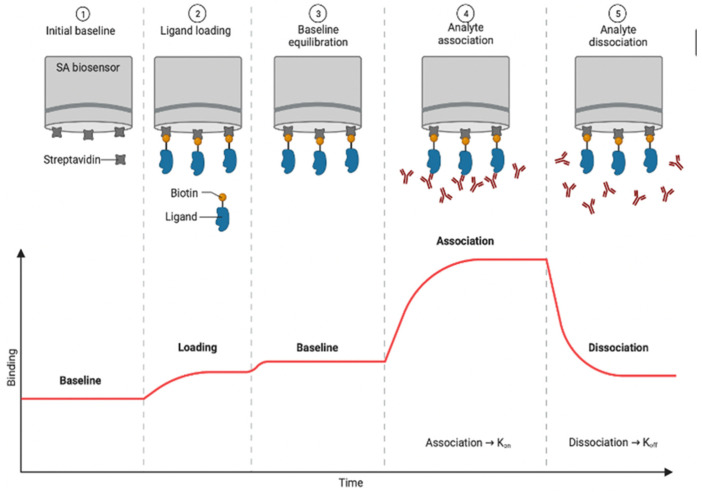
Experimental steps and representative (ideal) sensorgrams in a BLI assay based on Streptavidin fibers (SA biosensors) biosensors. Experimental steps showing the initial baseline in buffer (1), ligand loading using the streptavidin-biotin coupling method (2), baseline equilibration for drift correction in buffer (3), association with target analyte (4), and dissociation of ligand-analyte complex (5) (Figure created in BioRender (accessed on 5 May 2022)). Note that the loading plot shown here is for 100% surface coverage, which is not ideal for kinetic studies due to steric hindrance.

**Figure 5 biosensors-12-00938-f005:**
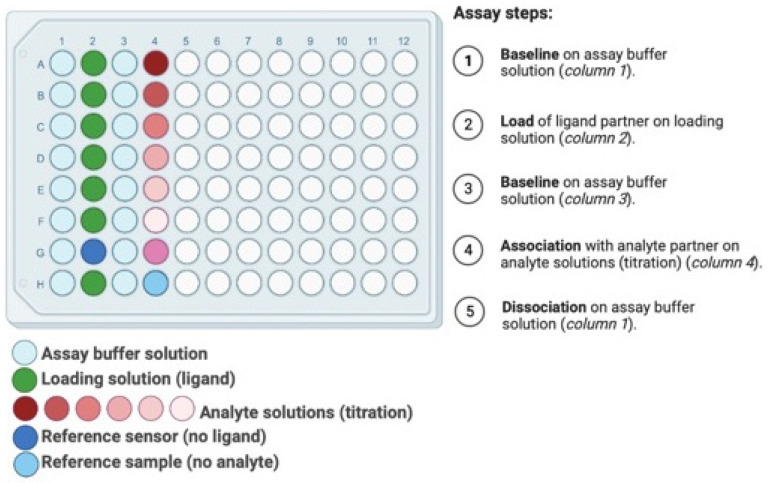
Plate layout for a typical BLI binding kinetics experimental plan in a 96 microwell format.

**Figure 6 biosensors-12-00938-f006:**
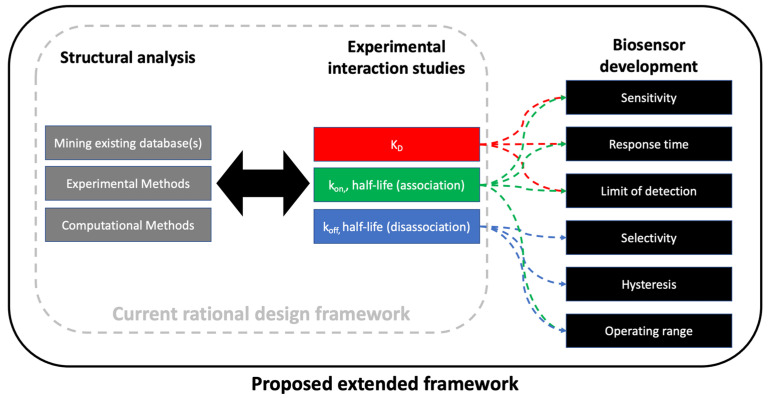
Framework for using structural predictions and receptor-target interaction studies (current rational design framework) to guide biosensor development. The three key domains of are structural analysis of ligand(s), experimental interaction studies, and biosensor development. These often occur in iteration, and the diagram is not intended to be restrictive by use of arrows. The proposed framework is outcome-based and focuses on six key performance indicators (KPI).

**Figure 7 biosensors-12-00938-f007:**
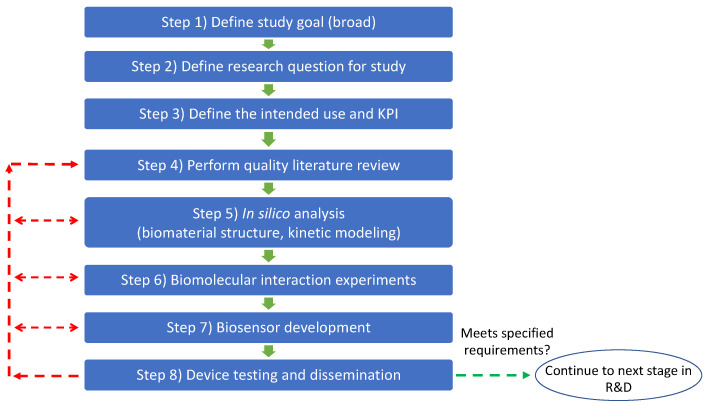
Flowchart for applying framework. Experimental design and iteration in steps 5–7 should always consider the study goal, intended use, and research question of the study as a guide for decision-making. This process also facilitates updating the quality literature review (use of PRISMA framework is highly suggested). Green arrows indicate that the prototype meets the requirements, and we can move to the next process step. Red arrows indicate that the prototype does not meet the specified requirements and an iteration to prior step(s) is required.

**Figure 9 biosensors-12-00938-f009:**
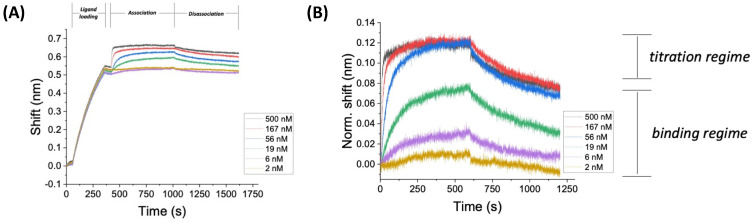
BLI sensorgrams for ACE2xDELTA in bicarbonate buffer at 30 °C. (**A**) Representative sensogram for ACE2xDELTA in bicarbonate buffer at 30 °C. Regions of ligand loading, analyte association, and disassociation are indicated at the top of the time series plot. The assay was conducted over concentrations that span the binding regime and the titration regime. (**B**) Association and dissociation curve for DELTA using [biot]ACE2 on SAV-coated fibers as the ligand in bicarbonate buffer at 30 °C.

**Figure 10 biosensors-12-00938-f010:**
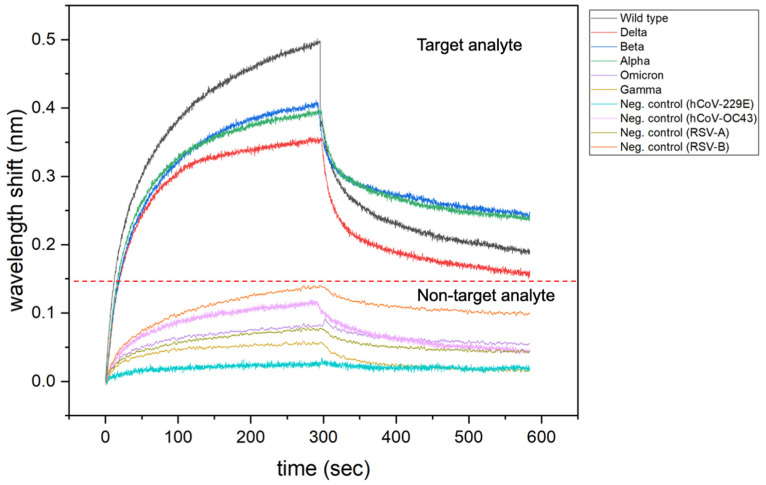
Representative association and dissociation curves for binding assay of ACE2 (ligand) towards target-analyte (SARS-CoV-2) and non-target analytes (Seasonal viruses).

**Figure 11 biosensors-12-00938-f011:**
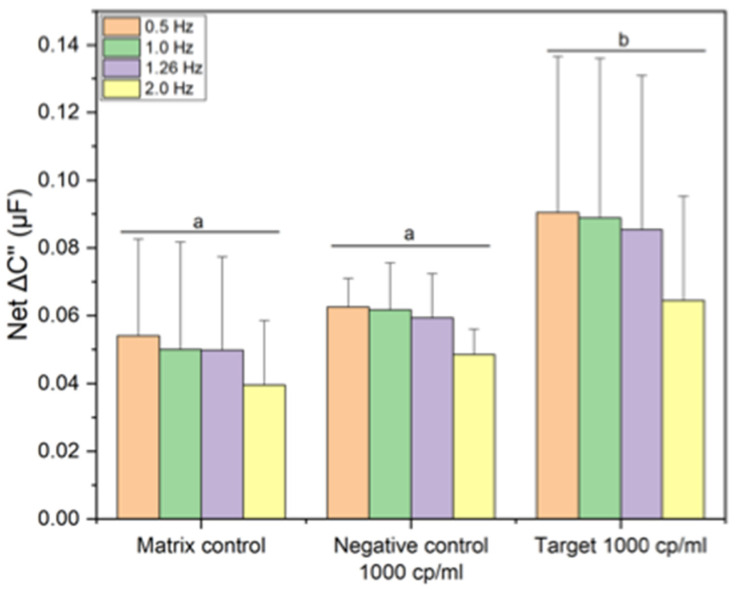
Representative capacitance response of ACE2-based biosensor for matrix effects (saliva target-free), negative control (common cold virus at 1000 copies/mL), and target (SARS-CoV-2 DELTA at 1000 copies/mL). Error bars represent standard deviation from the mean measurement (n = 3). Same letters represent groups with no significant difference (*p* < 0.05).

**Table 2 biosensors-12-00938-t002:** Advantages and limitations of optical and electrochemical biosensors.

Approach	Advantages	Limitations	References
Optical	Fast, sensitive, and high-throughput technique (96-well or 384-well format).	Limited by physics properties (mass).	[102,103,104]
	Rapid screening of molecules and optimum conditions for binding.	One mechanism of transduction.	
	Real-time qualitative monitoring of interactions (changes in wavelength shift).	Not suitable for small target-analyte since their mass may not generate a clear angle wave shift.	
	Allow the regeneration of the biosensors for reuse.	One immobilized ligand per biosensor tip.	
	Real-time and label-free detection.	Require costly equipment and laboratory structure.	
	Wide range of application, such as, quantitation, kinetics, isotyping of biomolecules.	The turbidity of biological samples might cause limitations on applying optical biosensors.	
		False-positive signals from any matrix artefactual.	
Electrochemical	Different electrochemical transduction signals can be employed (i.e., amperometric, potentiometric, voltammetric, impedimetric).	Limited by chemistry properties (redox signal).	[105,106]
	Sensible to small changes on the biosensor surface.	One-time use. Depending on the application, regeneration of the electrodes are not possible.	
	Miniaturization, Suitable for point-of-care, Scalability.		

## Data Availability

The original contributions presented in the study are included in the article/Appendix A; further inquiries can be directed to the corresponding author.

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
