# Peer review of "An Experimental Framework for Developing Point-of-Need Biosensors: Connecting Bio-Layer Interferometry and Electrochemical Impedance Spectroscopy"

_biosensors, 2022, doi:10.3390/bios12110938_

Round 1

Reviewer 1 Report

The article presents a framework to connect biomolecular interaction parameters with transducers for biosensor development and used an example based on the bio-layer interferometry for studying biorecognition interactions and an electrochemical impedance biosensor as transducer. Although the idea is interesting, several points should be re-considered before publication:

1. The title is misleading as the readers may think that this work is focused on the development of a novel bio-layer interferometric biosensor.

2. The purpose of the framework is not clear. Is it attempting to search the best combination of biorecognition element and transducer for a particular application? However, as shown in this manuscript, impedance biosensor was already chosen as the transducer, then how this framework is useful in biosensor development?

3. Section 4, third paragraph, an emerging class of biosensors based on localized surface plasmon resonance (LSPR) for characterizing ligand-analyte interactions is missing. A few commercial instruments based on LSPR are now on the market. Inclusion of LSPR biosensors and citation of these references are suggested: Langmuir 20 (2004) 8897-8902; Analytical Biochemistry 379 (2008) 1-7; Analytical Chemistry 84 (2011) 232-240; Lab on a Chip 12 (2012) 3882-3890; Analytical Chemistry 85 (2013) 245-250.

4. Figure 6 and Section 5, the performance indicators such as sensitivity, response time, limit of detection, and operation range depend a lot on the type of transducer, how the framework helps to assess these indicators? No results were presented in this aspect using the framework.

Reviewer 2 Report

This review/framework type of manuscript covers optical and electrochemical biosensors with specific focus on bio-layer interferometry as laboratory technique for studying biomolecular interactions. The manuscript is aimed at addressing the lack of detailed frameworks for implementing bio-layer interferometry in target studies for developing point-of-need biosensor tools and assay kits. Authors review and analyze connect information of molecular interaction studies in respect to key performance indicators of biosensors. 

The review is timely and valuable addition to the expanding biosensor area. Perhaps authors could also discuss how their proposed framework could be applied to other types of biosensors such as whole-cell biosensors. Some wider discussion/review of other type of biosensors could be beneficial to broaden the reader audience. 

Round 2

Reviewer 1 Report

The manuscript has been revised with consideration of the reviewers' comments. A minor point is that the title in the Supplementary Information had not been revised accordingly.

Author Response

Point 1: The manuscript has been revised with consideration of the reviewers' comments. A minor point is that the title in the Supplementary Information had not been revised accordingly. 

Response 1: Thank you for noticing this minor point. We modified the title in the Supplemental material. An updated version of the Supplemental material was attached.